# An Alliance of Polynitrogen Heterocycles: Novel Energetic Tetrazinedioxide-Hydroxytetrazole-Based Materials

**DOI:** 10.3390/molecules27185891

**Published:** 2022-09-11

**Authors:** Dmitry M. Bystrov, Alla N. Pivkina, Leonid L. Fershtat

**Affiliations:** 1N. D. Zelinsky Institute of Organic Chemistry, Russian Academy of Sciences 47 Leninsky Prosp., 119991 Moscow, Russia; 2N. N. Semenov Federal Research Center for Chemical Physics, Russian Academy of Sciences, 3 Kosygin Str., 119991 Moscow, Russia

**Keywords:** nitrogen heterocycles, energetic materials, tetrazine, hydroxytetrazole

## Abstract

Energetic materials constitute one of the most important subtypes of functional materials used for various applications. A promising approach for the construction of novel thermally stable high-energy materials is based on an assembly of polynitrogen biheterocyclic scaffolds. Herein, we report on the design and synthesis of a new series of high-nitrogen energetic salts comprising the C-C linked 6-aminotetrazinedioxide and hydroxytetrazole frameworks. Synthesized materials were thoroughly characterized by IR and multinuclear NMR spectroscopy, elemental analysis, single-crystal X-ray diffraction and differential scanning calorimetry. As a result of a vast amount of the formed intra- and intermolecular hydrogen bonds, prepared ammonium and amino-1,2,4-triazolium salts are thermally stable and have good densities of 1.75–1.78 g·cm^−3^. All synthesized compounds show high detonation performance, reaching that of benchmark RDX. At the same time, as compared to RDX, investigated salts are less friction sensitive due to the formed net of hydrogen bonds. Overall, reported functional materials represent a novel perspective subclass of secondary explosives and unveil further opportunities for an assembly of biheterocyclic next-generation energetic materials.

## 1. Introduction

A creation of novel functional organic materials remains one of the urgent goals in modern chemistry and materials science [1,2,3,4]. Such materials constitute a large variety of usually conjugated organic compounds with different chemical and physical properties. Recent achievements of numerous research groups worldwide confirmed that an incorporation of a nitrogen heteroaromatic motif usually enhances the quality of materials compared to their carbocyclic analogues [5,6,7]. In this regard, linear combinations of conjugated nitrogen heterocyclic moieties, especially of those mainly consisting of nitrogen atoms, demonstrate great application potential [8,9,10].

Among high-nitrogen heteroaromatic species, 1,2,4,5-tetrazine (six-membered ring with four nitrogen atoms) and tetrazole (five-membered ring with four nitrogen atoms) scaffolds retain leading positions in the chemistry community since materials derived thereof demonstrate improved functional properties. 1,2,4,5-Tetrazines may serve as components of photo- and electroactive materials [11,12], substrates for bioorthogonal processes [13,14] or precursors for diverse nitrogen heterocycles [15,16,17,18]. Tetrazoles are considered as carboxylic acid bioisosteres and are found in a wide range of pharmacological activity including some clinically approved pharmaceuticals [19,20,21]. Meanwhile, both tetrazine and tetrazole rings are used as paramount scaffolds in the construction of next-generation high-energy materials for mining, welding and other civil energetic applications [10,22]. As a rule, tetrazine- and tetrazole-based energy-rich compounds have a number of advantages including high nitrogen content, good thermal stability, acceptable sensitivity to mechanical stimuli and environmental compatibility [23,24]. A combination of C-C linked conjugated tetrazole and tetrazine rings afforded several thermally stable energetic materials (Figure 1), which, however, have low amounts of oxygen [25]. Meanwhile, oxygen balance defined as the degree to which an explosive can be oxidized is an important parameter for high-energy materials. Several strategies for an incorporation of oxygen-rich explosophoric moieties, such as trinitromethyl group [26,27] or furoxan ring [28,29], are commonly used to enhance the oxygen content. Unfortunately, these approaches inevitably entail a decrease in thermal stability and an increase in mechanical sensitivity. Therefore, a compromise between these criteria still remains an urgent task and defines future trends in materials science.

Recent investigations demonstrated the utility of the N-oxide functionalization strategy to balance physicochemical properties, mechanical sensitivity and oxygen balance of energetic materials [30,31,32,33]. Importantly, N-oxide functionality not only increases oxygen balance, but also allows for better crystal packing, and efficiently enhances detonation performance. In the case of 1,2,4,5-tetrazine, a preparation of its mono- and dioxide derivatives with promising energetic properties was reported [34]. For the tetrazole ring, an installation of the N-oxide moiety is complicated due to the azole nature of the heterocycle and involvement of nitrogen lone pairs into ring conjugation. A solution to this issue may comprise the formation of the hydroxytetrazole motif, which is also capable of the formation of energy-rich salts due to high acidity of the OH-group [35,36]. In this regard, an alliance of the tetrazinedioxide and hydroxytetrazole scaffolds may contribute advantageously from both heterocycles in terms of thermal stability and mechanical sensitivity and provide an evolutionary step toward functional organic materials of the future. Herein, we report on the design and synthesis of a new series of high-nitrogen energetic salts comprising the C-C linked 6-aminotetrazinedioxide and hydroxytetrazole frameworks (Figure 1). The presence of the amino group is desirable in terms of intra- and intermolecular hydrogen bonds formed between amino group hydrogens and N-oxide oxygens, which contribute to the density and stability of target materials. Complex multidisciplinary investigation of the thus-prepared compounds reveals a balanced set of their physicochemical and detonation parameters enabling their ability to replace existing explosives (e.g., RDX).

## 2. Results and Discussion

For the synthesis of target energetic materials, we decided to use 6-amino-3-cyanotetrazine **1** as a starting compound, since the nitrile group can be easily converted to the hydroxytetrazole motif [35,36]. Thus, our research was started from the optimization of the reaction conditions of nucleophilic substitution of the dimethylpyrazolyl fragment in a readily available 3-amino-6-(3,5-dimethylpyrazol-1-yl)-1,2,4,5-tetrazine [37] **2**. Several different solvents as well as cyanide sources were screened, and the results are summarized in Table 1. It was found that the source and the concentration of cyanide anions were crucial for the reaction to proceed. TMSCN was ineffective (entry 1), while acetone cyanohydrin provided target tetrazine **1** in 31–71% yields depending on the reagent excess and additives used (entries 2–4). A combination of KCN in hexafluoroisopropanol (HFIP) did not result in the formation of compound **1** (entry 5), but a replacement of HFIP with MeCN or DMF provided cyanotetrazine **1** in moderate yields (entries 6–8). It was also found that low water content proved to be essential for high yields of 3-amino-6-cyanotetrazine **1** as can be seen from entries 8–10; thus, the best results were achieved using dry DMF with an addition of molecular sieves under inert atmosphere (entry 9). We consider that any water present in the reaction mixture reacts with KCN to form HO^-^, which not only easily displaces dimethylpyrazolyl fragments to form a corresponding hydroxytetrazine derivative, but also induces the hydrolysis of the cyano group in the already formed product. It is also important that an excess of KCN can cause product hydrolysis upon aqueous work-up. To restrain this issue, the reaction mixture was poured into a slightly acidic aqueous ammonium chloride solution.

Curiously, an introduction of bis(dimethylpyrazolyl)tetrazine **3** in the same reaction under optimized conditions resulted again in a formation of 3-amino-6-cyano-1,2,4,5-tetrazine **1**. The great outcome was that not only was the yield of the target compound higher, but also the reaction time was reduced from 5 to 1.5 h. We supposed that 3-cyano-6-(3,5-dimethylpyrazol-1-yl)-1,2,4,5-tetrazine **4** formed initially. When the reaction mixture was poured into the ammonium chloride solution, the excess KCN neutralized the ammonium cation to form free ammonia, which then quickly displaced a second dimethylpyrazolyl fragment with the formation of 3-amino-6-cyano derivative **1** (Figure 1). From the technological point of view, the utilization of substrate **3** is more convenient and cost-effective since aminotetrazine **2** is synthesized from compound **3** [37]. Therefore, direct preparation of 3-amino-6-cyanotetrazine **1** from bis(dimethylpyrazolyl)tetrazine **3** allows to omit one reaction step.

With a developed procedure for the synthesis of 3-amino-6-cyanotetrazine **1** in hand, we performed stepwise functionalization to install the hydroxytetrazole scaffold to the tetrazine backbone. At first, compound **1** was oxidized to the corresponding di-N-oxide **5** using peroxytrifluoroacetic acid generated in situ from 85% H_2_O_2_ and trifluoroacetic anhydride (TFAA). Addition of hydroxylamine to the thus-obtained di-N-oxide **5** occurred easily, providing amidoxime **6** with almost quantitative yield. The latter was subjected to diazotization in HCl to form the corresponding chloroxime **7**. The chlorine atom in **7** can be easily substituted with an azide anion to form azidooxime **8** with an excellent yield. Acid-induced cyclization of azidooxime functionality in **8** furnished the formation of the hydroxytetrazole **9** (Figure 2). The overall yield of the target hydroxytetrazole **9** is remarkably high: 74% over five reaction steps starting from 3-amino-6-cyanotetrazine **1**.

Upon treatment with nitrogen-rich bases, hydroxytetrazole **9** was converted to the corresponding salts **10**–**12** in quantitative yields (Figure 3). Ammonia, hydroxylamine and 4-amino-1,2,4-triazole were used as commercially available and convenient base counterparts.

The structures of all synthesized compounds were confirmed by IR, ^1^H and ^13^C NMR spectroscopy as well as by elemental analysis. The structure of aminotriazolium salt **12** was additionally confirmed by ^15^N NMR spectroscopy (Figure 2). The signals were assigned on the basis of the literature values of resonance peaks in similar compounds. The tetrazinedi-N-oxide motif is symmetric due to aromaticity; therefore, there are only two nitrogen signals attributable to the tetrazine ring [34]. The N4 and N5 signals are more upfield (−90.4 ppm) relative to N1 and N3 (−80.3 ppm). On the contrary, the hydroxytetrazole motif is asymmetric, which is clearly shown by the presence of four signals similar to the previously reported data [38]. 1,2,4-Triazole fragment is symmetric and is shown by two nitrogen signals (−89.9 ppm for N12 and −194.5 ppm for N10, N11) [39]. Both amino groups are located close to −300 ppm.

The structure of salt **11** was further confirmed by X-ray diffraction study of a crystallohydrate grown from a methanol–water (1:1) mixture (Figure 3). Compound **11** crystallizes as a monohydrate in the monoclinic space group P2_1_/n with four formula units (4 anions, 4 cations, 4 water molecules) per cell and a density of 1.852 g·cm^−3^ at 100 K (Figure 3). The average length of CN and NN bonds in the tetrazinedi-N-oxide fragment is 1.347 Å, which is slightly above the values reported for similar 3,6-disubstituted tetrazine rings (1.335–1.344 Å). The average length of the N-oxide bond (1.270 Å) is also among the highest values (1.259–1.271 Å) reported to date [34,40,41].

The amino group and the tetrazine ring are nearly coplanar, which is shown by the torsion angle H(91)-N(9)-C(3)-N(6) = 2.86°. However, there is a noticeable twist between the planes of tetrazine and hydroxytetrazole rings, supported by the torsion angle N(8)-C(2)-C(1)-N(1) = 11.07°. This fact can be attributed to multiple hydrogen bonds formed by the hydroxytetrazole fragment.

The anionic units are stacked into infinite columns, which are supported by hydrogen bonds formed between exocyclic oxygen and ring nitrogen atoms of the anion and the surrounding water molecules and the hydroxylammonium counter-ions (Figure 4).

Hydroxytetrazole provides several H-bonds with water molecules: the first one is a moderate bond with O(1) (1.916 Å), and the second one formed with cyclic N(4) is weaker (2.065 Å). Additionally, each NH_2_ group interacts with an oxygen atom of the hydroxytetrazole fragment of the neighbor molecule and O(5) of the water molecule via two moderate H-bonds (1.979 and 1.971 Å, respectively; Figure 5).

Contacts between parallel anion stacks are provided by two hydroxylammonium cations, which are interconnected head-to-tail between each other with two equal H-bonds (1.983 Å). These dimeric spacers are then linked to anionic units through hydroxylammonium OH^-^ and NH_3_^+^ groups with two rather strong H-bonds (1.809 and 1.804 Å, respectively; Figure 6).

The surrounding of the N-oxides is different: O(3) forms two moderate bonds with a water molecule and a hydroxylammonium cation, and O(2) only forms a weak H-bond with water (2.872 Å).This fact results in a slight difference between lengths of these two bonds in crystals: the N(7)-O(3) bond is longer than the N(6)-O(2) bond (1.283 and 1.256 Å, respectively). As a result of vast amount of the formed H-bonds, the density of the monohydrate is relatively high (1.852 g·cm^−3^) and even higher than that for the water-free salt (1.78 g·cm^−3^).

The physical and detonation properties, such as thermal stability, density, enthalpy of formation, detonation performance, as well as sensitivity of all target compounds, were investigated. The results are summarized in Table 2. With an exception of hydroxylammonium salt **11**, analyzed compounds have acceptable thermal stability: for **10** and **12**, the extrapolated onset of the decomposition peak by DSC is above 200 °C. Measured densities fall in the range of 1.75–1.78 g·cm^−3^, which is quite good for organic energetic salts. Compounds **10**–**12** store large amounts of nitrogen (>56%), much more than that of benchmark nitramine energetic material RDX (37.8%). High nitrogen content should result in more eco-friendly reaction products. Combined nitrogen–oxygen contents of salts **10**–**12** are similar to that of RDX, while their oxygen balance to CO is slightly negative. Due to the presence of two additional carbon atoms in the amino-1,2,4-triazolium cation, the oxygen balance of salt **12** is the most negative in the presented series of energetic salts. At the same time, energetic materials **10**–**12** have high calculated enthalpies of formation within 413–780 kJ·mol^−1^. The enthalpy of formation of salt **12** is the highest since the contribution of the heteroaromatic amino-1,2,4-triazolium cation to the resulting value is the most significant in comparison with ammonium and hydroxylammonium cations. Having the enthalpies of formation and experimental densities in hand, we calculated the detonation velocities (D) and pressures (P) using the empirical equations included in the PILEM application [42]. All synthesized compounds show the high detonation performance, reaching that of benchmark RDX. As compared to the RDX, investigated salts are less friction sensitive.

## 3. Materials and Methods

**CAUTION!** Although we encountered no difficulties during the preparation and handling of compounds described in this paper, they are potentially explosive energetic materials that are sensitive to impact and friction. Mechanical actions of these energetic materials, involving scratching or scraping, must be avoided. Any manipulations must be carried out by using appropriate standard safety precautions.

### 3.1. General Methods

All reactions were carried out in well-cleaned oven-dried glassware with magnetic stirring. ^1^H and ^13^C NMR spectra were recorded on a Bruker AM-300 (300.13 and 75.47 MHz, respectively) spectrometer and referenced to residual solvent peak. ^15^N NMR spectrum was recorded on a Bruker AV-600 instrument (the frequency for ^15^N was 50.7 MHz) at room temperature. The chemical shifts are reported in ppm (δ). The IR spectra were recorded on a Bruker “Alpha” spectrometer in the range 400–4000 cm^−1^ (resolution 2 cm^−1^). Elemental analyses were performed by the CHN Analyzer Perkin-Elmer 2400. All solvents were purified and dried using standard methods prior to use. All standard reagents were purchased from Aldrich or Acros Organics and used without further purification.

### 3.2. X-ray Crystallography

X-ray diffraction data were collected at 100 K on a four-circle Rigaku Synergy S diffractometer equipped with a HyPix600HE area-detector (kappa geometry, shutterless ω-scan technique), using graphite monochromatized Cu K_α_-radiation. The intensity data were integrated and corrected for absorption and decay by the CrysAlisPro program [43]. The structure was solved by direct methods using SHELXT [44] and refined on *F*^2^ using SHELXL-2018 [45] in the OLEX2 program [46]. All non-hydrogen atoms were refined with individual anisotropic displacement parameters. The locations of hydrogen atoms H4, H51, H52, H91, H92, H101, H102 and H103 were found from the electron density-difference map; these hydrogen atoms were refined with individual isotropic displacement parameters. All other hydrogen atoms were placed in ideal calculated positions and refined as riding atoms with relative isotropic displacement parameters (for details, see Appendix A). CCDC 2169314 contains the supplementary crystallographic data for **11**. These data can be obtained free of charge via http://www.ccdc.cam.ac.uk/conts/retrieving.html (accessed on 17 July 2022)(or from the CCDC, 12 Union Road, Cambridge, CB21EZ, UK; or deposit@ccdc.cam.ac.uk).

### 3.3. Computational Methods

All calculations were performed with Gaussian09 software package [47]. The enthalpies of formation in the gas phase for all cases were calculated using the CBS-4M method [48]. The enthalpies of formation of salts in the solid phase were estimated on the basis of the crystal packing modeling method. Values for Δ_f_*H*° (atoms) were taken from the NIST database.
Δ_f_*H*°_(g, 298)_ = *H*_(Molecule, 298)_ − ∑*H*°_(Atoms, 298)_ + ∑Δ_f_*H*°_(Atoms, 298)_

Geometric optimization of all structures for crystal packing calculation was carried out using the DFT/B3LYP functional and the aug-cc-PVDZ basis set with a Grimme’s D2 dispersion correction [49]. The optimized structures were conformed to be true local energy minima on the potential-energy surface by frequency analyses at the same level.

In the calculation of lattice energy, the molecules were treated as rigid bodies with fixed point groups. We applied pairwise atom–atom potentials to describe the van der Waals and electrostatic point charges for Coulomb components of intermolecular energy. At the initial stage, “6–12” Lennard-Jones (LJ)-type potential parameters were used [50]. The electrostatic energy was calculated with a set of displaced point charge sites by program FitMEP [51]. The lattice energy simulations were performed with the program PMC [52].

It is well known that the majority of organic crystal structures studied experimentally belong to a rather limited number of space groups [53]. For brief assessment of crystal packing, we obtained the following ordered list of the most likely structural classes: P2_1_/c, P2_1_2_1_2_1_, P-1, Pca2_1_ and P1 with two independent molecules in cell, which cover more than 80% of the whole number of crystal structures in total [53]. Taking into account low deviation in the crystal lattice energies of polymorphs, such a calculation is considered reasonable.

Enthalpy of sublimation for **9** was calculated by the formula:ΔH_subl_ = −E_lat_ − 2RT
where R is the universal gas constant, E_lat_ is the lattice energy, T is temperature (298 K).

The new approach for salts proposes a technique based on modeling the crystal packing for a salt and a similar neutral compound (quasi-salt, cocrystal). The enthalpy of formation in this case is calculated as the average value between these two structures [54].

Detonation performance parameters (detonation velocity at maximal density and Chapman–Jouguet pressure) were calculated by a recently suggested set of empirical methods from PILEM application [42]. Note that the accuracy of the utilized PILEM empirical methods is comparable to benchmark thermodynamic code EXPLO5.

### 3.4. Thermal Analysis and Sensitivity Measurements

Thermal analysis was performed using the STA 449 F3 (Netzsch) apparatus. Samples of 0.5–1 mg mass were poured into alumina pans covered with pierced lids and heated to 600 °C with a constant rate of 5 K min^−1^. Impact sensitivity tests were performed using BAM-type machine according to STANAG 4489 [55]. Friction sensitivity was evaluated in agreement with STANAG 4487 [56]. The reported values correspond to 50% probability of explosions; other details can be found elsewhere [57].

### 3.5. Synthetic Procedures

**3-Amino-6-cyano-1,2,4,5-tetrazine (1).***Method A from 3-amino-6-(3,5-dimethyl-1H-pyrazol-1-yl)-1,2,4,5-tetrazine **2***: In a Schlenk flask under an argon atmosphere, 3-amino-6-(3,5-dimethyl-1*H*-pyrazol-1-yl)-1,2,4,5-tetrazine **2** (382 mg, 2 mmol), KCN (260 mg, 4 mmol) and oven-dried 3Ǻ molecular sieves (300 mg) were mixed, and dry DMF (9 mL) was added. The reaction mixture was stirred at ambient temperature for 5.5 h, poured into a solution of NH_4_Cl (16 g) in 200 mL of cold water and extracted with EtOAc (7 × 60 mL). The combined extracts were dried over MgSO_4_ and evaporated at reduced pressure. The resulting crude solid was purified by flash chromatography on SiO_2_ (eluent CH_2_Cl_2_-EtOAc, 4:1) yielding 206 mg (84%) of pure product. *Method B from 3,6-bis(3,5-dimethyl-1H-pyrazol-1-yl)-1,2,4,5-tetrazine **3***: In a Schlenk flask under an argon atmosphere, 3,6-bis(3,5-dimethyl-1*H*-pyrazol-1-yl)-1,2,4,5-tetrazine **3** (540 mg, 2 mmol), KCN (520 mg, 8 mmol) and oven-dried 3Ǻ molecular sieves (400 mg) were mixed, and dry DMF (15 mL) was added. The reaction mixture was stirred at ambient temperature for 1.5 h, poured into a solution of NH_4_Cl (12 g) in 150 mL of cold water and extracted with EtOAc (5 × 60 mL). The combined extracts were dried over MgSO_4_ and evaporated at reduced pressure. The resulting crude solid was purified by flash chromatography on SiO_2_ (eluent CH_2_Cl_2_-EtOAc, 4:1) yielding 220 mg (90%) of pure product. Red crystalline solid. mp = 176–177 °C (dec). IR (KBr), ν: 3432, 3331, 2253, 1670, 1637, 1531, 1501, 1038, 970 cm^−1^. ^1^H NMR (300 MHz, DMSO-*d_6_*) δ_H_, ppm: 9.12 (s, 2H). ^13^C{^1^H} NMR (75.5 MHz, DMSO-*d_6_*) δ_C_, ppm: 160.9, 144.6, 115.6. Calcd. for C_3_H_2_N_6_ (%): C, 29.51; H, 1.65; N, 68.84. Found (%): C, 29.59; H, 1.59; N, 68.67.

**6-Amino-3-cyano-1,2,4,5-tetrazine 1,5-dioxide (5).** First, 85% H_2_O_2_ (4.5 mL) was slowly added to trifluoroacetic anhydride (12 mL), cooled on an ice bath, and the temperature was kept below 10 °C. Then, a solution of 3-amino-6-cyanotetrazine (2.44 g, 20 mmol) in MeCN (30 mL) was added in one portion. The reaction mixture was stirred at 25 °C for 2 h, poured into water (250 mL) and extracted with EtOAc (10 × 50 mL). The combined extracts were dried over MgSO_4_ and evaporated at reduced pressure yielding 2.68 g (87%) of pure product. Yellow solid. mp = 191–192 °C (dec). ^1^H NMR (300 MHz, DMSO-*d_6_*) δ_H_, ppm: 9.69 (s, 2H). ^13^C{^1^H} NMR (75.5 MHz, DMSO-*d_6_*) δ_C_, ppm: 150.1, 129.9, 112.5. Calcd. for C_3_H_2_N_6_O_2_ (%): C, 23.38; H, 1.31; N, 54.54. Found (%): C, 23.51; H, 1.19; N, 54.31.

**6-Amino-3-(amino(hydroximino)methyl)-1,2,4,5-tetrazine 1,5-dioxide (6)**. First, 50% aqueous NH_2_OH (1.22 mL, 19 mmol) was added dropwise to a suspension of 6-amino-3-cyano-1,2,4,5-tetrazine 1,5-dioxide **5** (2.54 g, 16.5 mmol) in ethanol (66 mL) at 0 °C under vigorous stirring. The reaction mixture was stirred at 0 °C for 10 min and then at ambient temperature for an additional 1 h. The resulting solid was filtered off, washed with EtOAc (30 mL) and dried in air. Yield 2.96 g (96%). Orange solid. T_d_ = 207–208 °C. IR (KBr), ν: 3459, 3403, 3353, 3111, 1685, 1633, 1502, 1402, 1319, 1092, 947 cm^−1^. ^1^H NMR (300 MHz, DMSO-*d_6_*) δ_H_, ppm: 10.32 (s, 1H), 8.71 (br. s, 2H), 5.79 (s, 2H). ^13^C{^1^H} NMR (75.5 MHz, DMSO-*d_6_*) δ_C_, ppm: 147.1, 145.5, 145.2. Calcd. for C_3_H_5_N_7_O_3_ (%): C, 19.26; H, 2.69; N, 52.40. Found (%): C, 19.35; H, 2.61; N, 52.28.

**6-Amino-3-(chloro(hydroxyimino)methyl)-1,2,4,5-tetrazine 1,5-dioxide (7)**. Conc. HCl (59 mL) was added to a suspension of amidoxime **6** (2.805 g, 15 mmol) in distilled water (48 mL) at 0 °C under vigorous stirring. Then, a solution of NaNO_2_ (1.24 g, 18 mmol) in distilled water (18 mL) was added dropwise at 0 °C. The reaction mixture was stirred for 3.5 h at 0 °C, poured into 500 mL of water and extracted with EtOAc (8 × 60 mL). The combined extracts were dried over MgSO_4_ and evaporated at reduced pressure yielding 3.01 g (97%) of pure chlorooxime. Yield 3.01 g (97%). Yellow solid. T_d_ = 207–208 °C. IR (KBr), ν: 3391, 3291, 3262, 1644, 1494, 1303, 1088, 1007, 900 cm^−1^. ^1^H NMR (300 MHz, DMSO-*d_6_*) δ_H_, ppm: 13.20 (s, 1H), 8.96 (s, 2H). ^13^C{^1^H} NMR (75.5 MHz, DMSO-*d_6_*) δ_C_, ppm: 147.3, 145.1, 129.8. Calcd. for C_3_H_3_N_6_O_3_Cl (%): C, 17.45; H, 1.46; N, 40.69. Found (%): C, 17.40; H, 1.49; N, 40.61.

**6-Amino-3-(azido(hydroxyimino)methyl)-1,2,4,5-tetrazine 1,5-dioxide (8).** Chloroxime **7** (1.55 g, 7.5 mmol) was added in one portion to a solution of NaN_3_ (975 mg, 15 mmol) in 40 mL of distilled water at 0 °C. The reaction mixture was stirred at 0 °C for 10 min and then at ambient temperature for an additional 3.5 h. Then, conc. HCl (660 μL) was added, the reaction mixture was stirred for 10 min, and the yellow product was filtered off, washed with water (10 mL) and dried in air. Additional amounts of product were obtained from the mother liquor, which was evaporated under reduced pressure. Then, THF (20 mL) was added to the residue, and insoluble NaCl was filtered off. The resulting solution containing target azidooxime was evaporated under reduced pressure. The obtained products were combined to yield 1.50 g (94%) of target compound **8**. Yellow solid. T_d_ = 185–186 °C. IR (KBr), ν: 3374, 3246, 2172, 2135, 1645, 1499, 1311, 1085, 1029, 926 cm^−1^. ^1^H NMR (300 MHz, DMSO-*d_6_*) δ_H_, ppm: 12.46 (s, 1H), 8.90 (s, 2H). ^13^C{^1^H} NMR (75.5 MHz, DMSO-*d_6_*) δ_C_, ppm: 147.4, 144.0, 136.6. Calcd. for C_3_H_3_N_9_O_3_ (%): C, 16.91; H, 1.42; N, 59.15. Found (%): C, 16.70; H, 1.49; N, 58.99.

**6-Amino-3-(1-hydroxy-1*H*-tetrazol-5-yl)-1,2,4,5-tetrazine 1,5-dioxide (9)**. Azidooxime **8** (1.70 g, 8 mmol) was dissolved in a 20% HCl solution in dioxane (25 mL). The reaction mixture was stirred for 5 h at ambient temperature, poured into 100 mL of distilled water and evaporated at reduced pressure at 45 °C, adding water several times to completely remove any residual HCl. Yield 1.65 g (97%). Yellow solid. T_d_ = 213–214 °C. IR (KBr), ν: 3547, 3499, 1639, 1503, 1326, 1109, 956 cm^−1^. ^1^H NMR (300 MHz, DMSO-*d_6_*) δ_H_, ppm: 9.20 (s, 2H). ^13^C{^1^H} NMR (75.5 MHz, DMSO-*d_6_*) δ_C_, ppm: 148.0, 141.5, 139.6. HRMS (ESI) Calcd. for: C_3_H_4_N_9_O_3_^+^: 214.0431; Found: 214.0441 [M+H]^+^. Calcd. for: C_3_H_3_N_9_O_3_Na^+^: 236.0251; Found: 236.0251 [M+Na]^+^.

**Ammonium salt of 6-amino-3-(1-hydroxy-1*H*-tetrazol-5-yl)-1,2,4,5-tetrazine 1,5-dioxide (10)**. Dry gaseous ammonia was bubbled through a solution of hydroxytetrazole **9** (1.065 g, 5 mmol) in 40 mL of dry THF, cooled to 0 °C, for 5 min. The reaction mixture was stirred for an additional 30 min at 0 °C. The formed red solid was filtered off, washed with THF (50 mL) and dried in air. Yield 1.13 g (98%). Red solid. T_d_ = 212 °C. IR (KBr), ν: 3302, 3208, 1613, 1418, 1309, 1114, 956 cm^−1^. ^1^H NMR (300 MHz, DMSO-*d_6_*) δ_H_, ppm: 6.40 (br. s, 6H). ^13^C{^1^H} NMR (75.5 MHz, DMSO-*d_6_*) δ_C_, ppm: 147.3, 140.9, 137.4. ^14^N NMR (21.7 MHz, DMSO-*d_6_*) δ_N_: −362.1. Calcd. for C_3_H_6_N_10_O_3_ (%): C, 15.66; H, 2.63; N, 60.86. Found (%): C, 15.81; H, 2.49; N, 60.59.

**Hydroxylammonium salt of 6-amino-3-(1-hydroxy-1*H*-tetrazol-5-yl)-1,2,4,5-tetrazine 1,5-dioxide (11)**. First, 50% aqueous NH_2_OH (313 µL, 5.1 mmol) was added dropwise to a solution of hydroxytetrazole **9** (1.065 g, 5 mmol) in 20 mL of dry THF at 0 °C under vigorous stirring. The reaction mixture was additionally stirred at 0 °C for 30 min. The formed yellow solid was filtered off, washed with THF (30 mL) and dried in air. Yield 1.48 g (94%). Yellow solid. T_d_ = 155 °C. IR (KBr), ν: 3229, 2950, 1651, 1502, 1309, 1112, 956 cm^−1^. ^1^H NMR (300 MHz, DMSO-*d_6_*) δ_H_: 9.90 (br. s, 6H). ^13^C{^1^H} NMR (75.5 MHz, DMSO-*d_6_*) δ_C_, ppm: 146.7, 142.9, 137.5. Calcd. for C_3_H_6_N_10_O_4_ (%): C, 14.64; H, 2.46; N, 56.91. Found (%): C, 14.42; H, 2.57; N, 56.70.

**4-Amino-1,2,4-triazolium salt of 6-amino-3-(1-hydroxy-1*H*-tetrazol-5-yl)-1,2,4,5-tetrazine 1,5-dioxide (12)**. A solution of 4-amino-1,2,4-triazole (378 mg, 4.5 mmol) in 2 mL of MeOH was added dropwise to a solution of hydroxytetrazole **9** (959 mg, 4.5 mmol) in 16 mL of dry THF at ambient temperature under vigorous stirring. The reaction mixture was additionally stirred at ambient temperature for 30 min. The formed yellow solid was filtered off, washed with THF (30 mL) and dried in air. Yield 1.31 mg (98%). Yellow solid. T_d_ = 206 °C. IR (KBr), ν: 3331, 3134, 1638, 1500, 1320, 1100, 960 cm^−1^. ^1^H NMR (300 MHz, DMSO-*d_6_*) δ_H_, ppm: 9.18 (s, 2H), 8.55 (s, 2H), 6.02 (br. s, 3H). ^13^C{^1^H} NMR (75.5 MHz, DMSO-*d_6_*) δ_C_, ppm: 147.8, 144.6, 140.9, 140.2. ^15^N NMR (50.7 MHz, DMSO-*d_6_*) δ_N_, ppm: −4.8, −14.3, −51.2, −80.3, −89.9, −90.4, −105.6, −194.5, −306.6, −312.5. Calcd. for C_5_H_7_N_13_O_3_ (%): C, 20.21; H, 2.37; N, 61.27. Found (%): C, 20.36; H, 2.29; N, 60.97.

## 4. Conclusions

In conclusion, a series of novel energetic organic salts comprising C-C bridged tetrazinedi-N-oxide and hydroxytetrazole rings and nitrogen-rich cations was synthesized starting from the parent compound 3-amino-6-cyano-1,2,4,5-tetrazine. These energetic materials were well characterized by IR and ^1^H, ^13^C, ^15^N NMR spectroscopy, elemental analysis and differential scanning calorimetry. The molecular structure of the hydroxylammonium salt **11** was additionally confirmed by single-crystal X-ray diffraction. The anionic units in energetic salt **11** are stacked into infinite columns, which are supported by hydrogen bonds formed between exocyclic oxygen and ring nitrogen atoms of the anion and the hydroxylammonium counter-ions. Synthesized energetic salts have high enthalpies of formation and excellent detonation performance, which together with high nitrogen content, make these compounds promising green alternatives for commonly used secondary explosive RDX. Moreover, reported high-energy salts have lower friction sensitivity compared to RDX, which additionally confirms their suitability for energetic applications as secondary explosives.

## Data Availability

Data obtained in this project are contained within this article and are available upon request from the authors.

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
