# Peer review of "An Alliance of Polynitrogen Heterocycles: Novel Energetic Tetrazinedioxide-Hydroxytetrazole-Based Materials"

_molecules, 2022, doi:10.3390/molecules27185891_

Round 1

Reviewer 1 Report

The authors report on a promising approach for the construction of novel thermally stable high-energy materials is based on an assembly of polynitrogen biheterocyclic scaffolds.

The work seems to have been carried out properly and completely.

Fig. 3: Selected bond lengths ((UNIT is missing))

Fig. 5 and 6:  unit is missing

Tab.2:  Which methods were used to calculate the VoD and pressure ?

Computational Methods:  Here the authors state "The enthalpies of formation of salts in the solid phase were estimated on the 252 basis of the crystal packing modeling method."  This is totally NOT clear.  Which models ? which method? references ?

Author Response

The authors are grateful to the reviewers for their valuable and positive comments on our manuscript. We did all our best to consider the issues raised and improved our manuscript accordingly.

Response to the Reviewer 1:

Fig. 3: Selected bond lengths ((UNIT is missing))

Fig. 5 and 6:  unit is missing

We were unable to add units directly onto the figures, but we added an appropriate phrase to the corresponding legends.

Tab.2:  Which methods were used to calculate the VoD and pressure?

Detonation performance parameters (detonation velocity at maximal density and Chapman-Jouguet pressure) were calculated by recently suggested set of empirical methods from PILEM application [ref. 42 in the manuscript]. This was added to the experimental section as well.

Computational Methods:  Here the authors state "The enthalpies of formation of salts in the solid phase were estimated on the 252 basis of the crystal packing modeling method."  This is totally NOT clear.  Which models ? which method? references ?

Indeed, we did not provide necessary details. In the revised version we provided the computational details for calculations of the enthalpies of formation with the corresponding references.

Reviewer 2 Report

The manuscript entitled "An Alliance of Polynitrogen Heterocycles: Novel Energetic Tetrazinedioxide-Hydroxytetrazole-Based Materials" by Bystrov et al. is a well-written and thoroughly studied manuscript. The synthesis of multiple new molecules and characterization of one of those by SCXRD makes this contribution significant. Therefore, I recommend publication with minor revision on following points.

1. When N-content% is almost equal for compound 10 and 12 the Oxygen balance for them is significantly different. What is this attributed to? author needs to emphasise.

2. The very high enthalpy of formation for the salt 12 should be discussed with explanantion?

3. In Supporting Information file, the important reference for the NMR solvent peaks have not been labelled in any spectrum. This should be performed.

Author Response

The authors are grateful to the reviewers for their valuable and positive comments on our manuscript. We did all our best to consider the issues raised and improved our manuscript accordingly.

Response to the Reviewer 2:

1. When N-content% is almost equal for compound 10 and 12 the Oxygen balance for them is significantly different. What is this attributed to? author needs to emphasise.

This fact is attributed to the presence of additional carbon atoms in the structure of amino-1,2,4-triazolium cation for salt 12. This explanation was added to the manuscript.

2. The very high enthalpy of formation for the salt 12 should be discussed with explanantion?

A contribution of amino-1,2,4-triazolium cation in the enthalpy of formation of salt 12 is much more significant than contributions from ammonium and hydroxylammonium cations. Thus, the enthalpy of formation for salt 12 is higher. A brief discussion was added to the manuscript.

3. In Supporting Information file, the important reference for the NMR solvent peaks have not been labelled in any spectrum. This should be performed.

The SI file was revised according to the recommendation.

Round 2

Reviewer 1 Report

The revised version is now fully acceptable for publication.